# PSMA PET-CT Imaging Predicts Treatment Progression in Men with Biochemically Recurrent Prostate Cancer—A Prospective Study of Men with 3 Year Follow Up

**DOI:** 10.3390/cancers14112717

**Published:** 2022-05-31

**Authors:** Sean Ong, Claire Pascoe, Brian D. Kelly, Zita Ballok, David Webb, Damien Bolton, Declan Murphy, Shomik Sengupta, Patrick Bowden, Nathan Lawrentschuk

**Affiliations:** 1EJ Whitten Prostate Cancer Research Centre, Epworth HealthCare, Richmond, VIC 3121, Australia; cea.pascoe@gmail.com (C.P.); webbs@bigpond.net.au (D.W.); damienmbolton@gmail.com (D.B.); declan.murphy@petermac.org (D.M.); pat.bowden@epworth.org.au (P.B.); lawrentschuk@gmail.com (N.L.); 2Young Urology Researcher’s Organisation, Melbourne, VIC 3000, Australia; 3Department of Surgery, University of Melbourne, Parkville, VIC 3010, Australia; 4Division of Cancer Surgery, Peter MacCallum Cancer Centre, Melbourne, VIC 3000, Australia; briandaniel.kelly@petermac.org; 5Department of Urology, Eastern Health, Box Hill, VIC 3128, Australia; shomik.sengupta@monash.edu; 6Department of Nuclear Medicine, Richmond Medical Imaging, Richmond, VIC 3121, Australia; zita.ballok@lumusimaging.com.au; 7Olivia Newton-John Cancer and Wellness Centre, Austin Health, Heidelberg, VIC 3084, Australia; 8Eastern Health Clinical School, Monash University, Box Hill, VIC 3128, Australia; 9Department of Urology, Royal Melbourne Hospital, Melbourne, VIC 3051, Australia

**Keywords:** PSMA PET-CT, prostate cancer, biochemical recurrence

## Abstract

**Simple Summary:**

Prostate-specific membrane antigen (PSMA) positron emission tomography-computed tomography (PET-CT) is an essential imaging tool that is used to locate prostate cancer when it recurs. The results of this scan are used to guide clinical decisions for the management of cancer. However, the long-term effect of these clinical decisions is yet to be determined. In this study, we followed men with recurrent prostate cancer for 3 years after they had a clinical decision made based on a PSMA PET-CT. Our results showed that 75% of men had no addition or change in their treatment plan 3 years after their initial clinical decision was made. In men with a PSMA PET-CT that showed no suspected cancer, 85% had no addition or change to their treatment plan. This indicates that clinical decisions made using PSMA PET-CT in this setting can have a medium- to long-lasting effect.

**Abstract:**

Prostate-specific membrane antigen (PSMA) positron emission tomography-computed tomography (PET-CT) is a novel imaging modality used to stage recurrent prostate cancer. It has the potential to improve prognostication and ultimately guide the timing of treatment for men with recurrent prostate cancer. This study aims to assess the clinical impact of PSMA PET-CT by analyzing its predictive value of treatment progression after 3 years of follow-up. In this prospective cohort study of 100 men, patients received a PSMA PET-CT for restaging of their disease which was used by a multi-disciplinary team to make a treatment decision. The primary endpoint was treatment progression. This was defined as the addition or change of any treatment modalities such as androgen deprivation therapy (ADT), radiation therapy or chemotherapy. The median follow-up time was 36 months (IQR 24–40 months). No treatment progression was found in 72 (75%) men and therefore 24 (25%) patients were found to have treatment progression. In men with a negative PSMA PET-CT result, 5/33 (15.1%) had treatment progression and 28/33 (84.8%) had no treatment progression. In conclusion, clinical decisions made with PSMA PET-CT results led to 75% of men having no treatment progression at 3 years of follow-up. In men with negative PSMA PET-CT results, this increased to 85% of men.

## 1. Introduction

New imaging modalities have made a significant impact on the way prostate cancer (PCa) can be characterized. Prostate-specific membrane antigen positron emission tomography-computed tomography (PSMA PET-CT) imaging is one such modality that is increasingly utilized. The proPSMA trial has demonstrated PSMA PET-CT to be more accurate than conventional imaging (CT and Bone Scan(BS)) when staging PCa [1]. However, the true clinical impact of this new modality at multiple stages of the diagnostic pathway for PCa is still being investigated.

PSMA is a non-soluble type 2 cell surface transmembrane glycoprotein that is expressed on the surface of endothelial cells [2]. Contrary to its name, PSMA is not specific to prostate tissue [3]. In fact, it has been seen in normal, inflammatory, benign and malignant tissues of the body [3,4]. Some studies theorize that PSMA has a pathophysiologic role in carcinogenesis, in particular in regard to folate regulation and tumor-specific neovascularization [5,6,7,8]. Specific to prostate cancer, PSMA is found in over 90% of PCa cells at expression rates that are 100–1000 times greater than physiological levels especially in higher grade cancers [9,10,11,12]. Thus, PSMA has now been targeted for diagnostic imaging and therapeutic advances.

When targeted by small ligands with an attached radio-labeled marker such as gallium (^68^Ga) or fluorine (^18^F), PSMA can be combined with PET technology to produce an image that detects the presence of PCa. PSMA PET-CT is currently recommended in men with biochemical recurrence (BCR) post radical prostatectomy or in men fit for curative salvage treatment after radiotherapy [13]. However, data are limited regarding the outcomes of consequent decisions after imaging [13].

A recent meta-analysis of mostly retrospective data has shown PSMA PET-CT to have a high specificity (99%) and relatively high sensitivity (75%) on a per lesion basis [14]. The same study also found positivity rates to be low when serum prostate-specific antigen (PSA) levels were less than 1 ng/mL [14]. Furthermore, the positive predictive value of PSMA PET-CT is likely to be lower in men with solitary lesions, especially in the ribs [15]. The clinical utility of PSMA PET-CT is therefore an important clinical question for ongoing studies as PSMA PET-CT becomes more available for use globally. The question remains of whether increased positivity rates in imaging translate to changed management and ultimately improved patient outcomes. Alternately, is a negative scan predictive of a more indolent course?

The natural history of men with BCR is highly variable. Currently, post-radiation therapy, BCR can be defined as a rise of >2 ng/mL above nadir [13]. However, the correct definition of BCR post-radical prostatectomy is less clear. The most current EAU-EANM-ESTRO-ESUR-ISUP-SIOG guideline state that after radical prostatectomy, there is no specific PSA threshold defining recurrence. They do, however, acknowledge that the best threshold for predicting further metastases is a rising PSA of >0.4 ng/mL [16]. The AUA-ASTRO-SUO guidelines state that BCR should be defined as two consecutive rising PSA readings >0.2 ng/mL [17]. Nevertheless, after the identification of BCR, prognostic markers including ISUP grade, pre-treatment PSA, PSA kinetics (doubling time and PSA velocity) and T-staging have been found to predict the likelihood of disease progression [18]. Consequently, risk stratification tools such as the D’Amico criteria [19], CAPRA score [20], and EAU risk groups [21] have been developed incorporating these variables. Early studies have shown that PSMA PET-CT may have a prognostic role [22]; however, to our knowledge, no stratification tools have included PSMA PET-CT.

Prognostication is ultimately important for determining the timing of salvage treatment. The challenge for clinicians lies in delaying the progression to metastatic disease without over-treating these men [23,24]. Results from the recent RAVES [25] and RADICALS [26] trials coupled with older salvage treatment studies [27] reveal a potential “sweet spot” for early salvage treatment between PSA 0.2 and 0.5 ng/mL for men with BCR post radical prostatectomy (RP). However, for men after radiotherapy or focal therapy, or for men with a persistent PSA after radical treatment, the timing of salvage treatment is less clear. PSMA PET-CT may add value to determining the timing of salvage treatment as well as identifying the optimal treatment modality to offer.

In this study, our primary aim was to investigate the outcomes of men with BCR who utilized PSMA PET-CT in their management pathway.

## 2. Materials and Methods

### 2.1. Patients

Patients were recruited through the seven campuses of Epworth HealthCare hospitals throughout Victoria, Australia, between March 2017 and March 2018. All patients signed an information and consent form to be recruited into the study.

### 2.2. Eligibility Criteria

Men were eligible for the study if they had BCR for re-staging. BCR was defined in accordance with the ASTRO-Phoenix consensus definition: prostate-specific antigen (PSA) 0.2 ng/mL at least 6 weeks post-prostatectomy or a PSA rise of 2 ng/mL or above nadir following radiotherapy [28]. Patients were recruited irrespective of conventional imaging findings.

Exclusion criteria were another active malignancy within the last 5 years, excluding non-melanoma skin cancer, men having had prostate bed radiation within 6 months of enrolment or men who were having androgen deprivation therapy started less than 6 months before enrolment.

### 2.3. Study Design

This study was a multi-center prospective cohort study of men with biochemically recurrent prostate cancer based on the inclusion and exclusion criteria mentioned above.

Patients were enrolled and consented to the study. Baseline demographic data were collected along with medical history regarding their prostate cancer diagnosis and prior treatment. These men then received a PSMA PET-CT scan as part of their cancer re-staging. Each case including the results of the PSMA PET-CT scan was then presented in a multidisciplinary meeting (MDM) including urologists, medical oncologists, radiation oncologists, uro-pathologists and uro-radiologists to determine a consensus management plan. Men received a minimum of 6 monthly PSAs for follow-up with imaging or further presentation to an MDM based on PSA results or clinical concerns. The need for treatment progression and type of treatment was discussed in the MDM consensus. Data pertaining to follow-up investigations and treatment additions or changes were prospectively collected until March 2021.

### 2.4. PSMA PET-CT Imaging and Radiotracer

PSMA PET-CT scans were performed at various radiographic centers around Victoria, Australia. All centers used ^68^Gallium-PSMA-11 as the radiotracer. These were produced using a standardized protocol which defined minimum specifications for radiopharmaceutical production. ^68^Gallium-PSMA-11 scanning protocols were similar across sites. PET-CT scanners acquired PET images from thighs to vertex at 60 min after administration of 2 MBq/kg body weight ± 5% of ^68^Gallium-PSMA-11. A low dose non-contrast CT was performed during tidal respiration for attenuation correction and anatomical correlation.

All PSMA PET-CT scans were reported by a radiologist at the center where they were performed and then reviewed by a urology-specific radiologist at a multi-disciplinary meeting.

### 2.5. Statistical Analysis

The primary endpoint of this study was treatment progression post initial management plan. This was defined as the addition or change of any treatment modalities such as androgen deprivation therapy (ADT), radiation therapy or chemotherapy. Disease progression seen on imaging or as a PSA progression with no addition or change in treatment modality was not classified as treatment progresses. Univariate and multivariate logistic regression analysis was performed. All statistical analysis was performed using the R software. A *p*-value of <0.05 was considered to be statistically significant.

## 3. Results

### 3.1. Patient Characteristics

In total, 100 men were recruited for this study. Four men were excluded from statistical analysis due to loss of follow-up. The mean age of the remaining 96 men was 71 years (interquartile range (IQR) 67.5–75.3 years) and the median PSA at the time of the PSMA PET-CT scan was 0.82 ng/mL (IQR 0.27–2.95 ng/mL). The ISUP grade of three men was unknown. Of the remaining patients, ISUP grades 1, 2, 3, 4 and 5 were confirmed in 4, 18, 37, 6 and 28 men, respectively. Previous treatment with radical prostatectomy (RP), brachytherapy or external beam radiation therapy (EBRT) only was received by 52, 5 and 4 men respectively. Salvage radiation therapy post RP was received by 22 men. Other treatment combinations previously received by the patients are outlined in Table 1.

Of the men who had initial treatment with RP, 30 men had ISUP grade 4 or 5 found in their RP specimen (1 patient’s ISUP score not found). Positive margins and intraductal carcinoma were found in 28 (6 not reported) and 17 (13 not reported) RP specimens, respectively. Positive lymph nodes were found in 13 specimens; however, 23 men did not receive a lymph node dissection. Stage T2, T3a, T3b and T4 were found in 19, 37, 31, and 1 men, respectively (5 were not reported).

### 3.2. PSMA PET-CT Scan Results

All men underwent a ^68^Ga-PSMA-11 PET scan. Results showed no suspicious avid lesions in 33 (34.3%) men. A locally avid prostate recurrence was seen on PSMA PET-CT in six (6.25%) men. Pelvic lymph nodes and extra-pelvic lymph nodes were seen in 37 (38.5%) and 14 (14.6%) scans, respectively. Bony lesions and distal organ lesions were seen in 12 (12.5%) and 7 (7.29%) scans, respectively.

Results stratified by PSA level showed that 3/6 (50%) men with PSA <0.2 ng/mL had a PSMA PET-CT positive result. For men with PSA levels between 0.2–0.5 ng/mL and 0.51–1 ng/mL, 14/33 (42.4%) and 11/14 (78.6%) had positive results respectively. For men with PSA levels between 1 and 5 ng/mL and over 5 ng/mL, 15/22 (68.2%) and 20/21 (95.2%) had positive results respectively. These results are shown in Table 2 and Table 3.

### 3.3. Management Decisions

Management decisions post-diagnostic workup included pelvic radiotherapy, stereotactic body radiation therapy (SRBT), salvage prostatectomy, ADT only, chemotherapy, SRBT and pelvic radiotherapy and surveillance for 11, 15, 5, 16, 12, 35 and 2 men, respectively. This is shown in Table 4.

### 3.4. Follow-Up and Treatment Progression

Four (4%) men were lost to follow-up. Of the 96 remaining men, the median follow-up time was 36 months (IQR 24–40 months). No treatment progression was found in 72 (75%) men and therefore 24 (25%) patients were found to have treatment progression. Disease progression on imaging or PSA levels with no treatment progression was seen in seven (7.3%) men. Progression to ADT, radiotherapy or chemotherapy was seen in 15, 6, and 10 men, respectively. This is shown in Table 5.

### 3.5. PSMA PET-CT as Predictor of Treatment Progression

In men with a negative PSMA PET-CT result, 5/33 (15.1%) had treatment progression and 28/33 (84.8%) had no treatment progression. Of the patients who had no treatment progression, 5/28 (17.9%) had a rise in PSA or disease seen on imaging but no addition or change in treatment.

In men with positive PSMA PET-CT results, 19/63 (30.1.1%) had treatment progression and 44/63 (69.8%) had no treatment progression. Of the patients who had no treatment progression, 2/63 (3.2%) had a rise in PSA or disease seen on imaging but no addition or change in treatment. This is shown in Table 6.

Additionally, on multivariate analysis (Table 7), age, Gleason grade group, PSA at time of PSMA PET and presence of a solitary pelvic node were not found to be predictors of a change in management. However, a positive PSMA PET scan for metastatic disease was a significant predictor of the initiation of a change in a patient’s management.

## 4. Discussion

PSMA PET-CT is a new imaging modality currently being utilized for the detection of prostate cancer in men. For men with recurrent prostate cancer, PSMA PET-CT has now been incorporated into guidelines as a restaging scan when they have BCR [13]. This, however, is largely based on retrospective studies which have shown increased positivity rates of PSMA PET-CT compared to CT; BS, however, still shows suboptimal rates of detection, especially at lower PSA levels [14]. Despite this, studies have shown that PSMA PET-CT results will impact clinical decision-making for these men [29]. It is therefore important to investigate the outcomes of these clinical decisions. This study reports medium-long term treatment progression results in men with BCR triaged by PSMA PET-CT.

In this multi-site prospective cohort study of 100 men, we found that clinical decisions based on PSMA PET-CT findings led to 75% of men having no treatment progression at 3 years of follow-up. Comparably, 3-year outcomes from a study by Emmett et al. found that 64.5% of men were free from progression post salvage radiotherapy [22]. The 10% difference in results could be explained by their definition of progression which included a rise in PSA (defined as a rise of serum >0.2 ng/mL above nadir, with the addition of any ADT or radiotherapy) and the specific cohort of men they analyzed. Our results illustrate that for men with BCR, triaging with PSMA PET-CT culminates in clinical decisions that have low rates of treatment progression regardless of pre- and post-BCR treatments.

In regard to men with negative PSMA PET-CT scans, we found 84.5% had no treatment progression at 3 years. This is likely due to the image representing a low volume of disease and low PSMA expression. Interestingly, 5/23 (17.9%) of these men had disease progression seen on imaging or serum PSA; however, no treatment progression was seen. Our results again align with the results of Emmett et al. [22] who found a negative PSMA PET-CT to be an important predictor of disease control. As guidelines are being created for the timing of treatment and follow-up of men with BCR, our results should provoke further prospective studies incorporating the use of PSMA PET-CT as a biomarker for initiation or delay of treatment. Specifically for men considering salvage radiotherapy, although a “sweet spot” for treatment between PSA 0.2 and 0.5 ng/mL may exist, perhaps a negative PSMA PET-CT can improve risk stratification to avoid over-treatment of potentially indolent disease. Furthermore, the issue of timing of subsequent PSMA PET-CT after an initial negative scan needs to be addressed.

One thing to consider is the possibility of prostate cancer showing androgen receptor-independent growth. With the development of ADT in recent years, the number of these prostate cancers is growing. This has implications for subtypes such as neuroendocrine-differentiated or small cell prostate cancer which are known to be PSMA PET-CT negative but tend to be more lethal [30]. In these cases, clinicians need to be mindful of the negative PSMA PET-CT and consider the utilization of conventional imaging modalities (CT, WBBS, MRI) [31], novel imaging techniques (Ga-68-DOTATOC PET-CT or DLL3 PET-CT) [32] or measurement of serum neuron-specific enolase [33].

Lastly, our results show a myriad of treatment decisions made based on PSMA PET-CT findings. In general, men post radical treatment only were offered salvage prostatectomy or radiotherapy if no targets outside the prostate were seen. Men were offered metastasis-directed treatment if feasible or systemic therapy for a high volume of disease outside the prostate. We found, however, that the majority of men (35 men) opted for surveillance with PSA with only a small number of men receiving salvage prostatectomy (5 men) or salvage radiation (11 men). This is likely due to management plans being created through a shared decision-making process based on guidelines, expert opinion, patient preference and availability.

The main limitation of this study in our opinion is the lack of conventional imaging and MRI for all patients. Given this, men that received conventional imaging or MRI may have had added information factored into their clinical decision making. Additionally, patient preference was factored into the clinical decision-making process for all men. However, we believe this is representative of a real-world scenario. Secondly, at the time of conception of this study, the definition for BCR after RP in guidelines was >0.2 ng/mL; however, the latest EAU guidelines state that the threshold that predicts further metastases is a PSA >0.4 ng/mL and rising. Therefore, this cohort included men who may have had a lower risk of disease progression. Lastly, this was a smaller study of only 100 men. Larger, high-powered trials are needed to further investigate the role of PSMA PET-CT in this space.

## 5. Conclusions

In this multicenter, prospective cohort study of men with BCR, clinical decisions made with PSMA PET-CT results led to 75% of men having no treatment progression at 3 years follow-up. In men with negative PSMA PET-CT results, a higher rate of men had no treatment progression. This should provoke further prospective studies regarding the timing of PSMA PET-CT and initiation of treatment for men with recurrent prostate cancer.

## Figures and Tables

**Table 1 cancers-14-02717-t001:** Patient characteristics.

Number of patients recruited		100
Number of patients included in analysis		96
Mean age		71
Median PSA at time of PSMA PET-CT		0.82 (IQR 0.271–2.95)
ISUP grade	Unknown	3
	ISUP grade 1	4
	ISUP grade 2	18
	ISUP grade 3	37
	ISUP grade 4	6
	ISUP grade 5	28
Initial management	RP only	52
	EBRT only	5
	Brachytherapy	4
	Brachytherapy + CT	2
	RP + salvage RT	22
	RP + SBRT + CT	4
	RP + RT + CT	3
	RP + CT	2
	ADT only	2

RP—radical prostatectomy; EBRT—external beam radiotherapy; RT—radiotherapy; SBRT—stereotactic body radiation therapy; CT—chemotherapy; ADT—androgen deprivation therapy.

**Table 2 cancers-14-02717-t002:** PSMA PET-CT results.

Positive	63
Negative	33
Prostate lesion only	6
Pelvic lymph nodes	37
Extra-pelvic lymph nodes	14
Bony lesion	12
Distal organ lesion	7

**Table 3 cancers-14-02717-t003:** PSMA PET-CT results stratified by PSA level.

PSA Level (ng/mL)	PSMA PET-CT Positive	PSMA Negative
<0.2	3 (50%)	3 (50%)
0.2–0.5	14 (42.4%)	19 (57.6%)
0.51–1	11 (78.6%)	3 (21.4%)
1–5	15 (68.2%)	7 (31.8%)
>5	20 (95.2%)	1 (4.8%)

**Table 4 cancers-14-02717-t004:** Clinical decisions made using PSMA PET-CT results.

Pelvic RTx	11
SBRT	15
Salvage prostatectomy	5
ADT only	16
Chemotherapy	12
Surveillance	35
SBRT + Pelvic radiation	2

RT—radiotherapy; SBRT—stereotactic body radiation therapy; ADT—androgen deprivation therapy.

**Table 5 cancers-14-02717-t005:** Follow-up and treatment progression.

Follow-up	Lost to follow-up	4
	Median follow-up	36 months (IQR 24–40 months)
Treatment progression	No treatment progression	65 patients
	Treatment progression	24 patients
	Progression on imaging or PSA but no treatment progression	7
	Progressed to ADT	15
	Progressed to Radiotherapy	6
	Progressed to Chemotherapy	10

**Table 6 cancers-14-02717-t006:** PSMA PET-CT as a predictor of treatment progression.

PSMA PET-CT negative	Treatment progression	5 men
	No treatment progression	28 men
	Progression on imaging or PSA but no treatment progression	5 men
PSMA PET-CT positive	Treatment progression	19 men
	No treatment progression	44 men
	Progression on imaging or PSA but no treatment progression	2 men

**Table 7 cancers-14-02717-t007:** Multivariate analysis of factors predicting a change in management.

Predictors	Odds Ratios	Confidence Intervals	*p*-Value
Age	1.06	0.97–1.16	0.207
Grade group of initial prostate biopsy	1.34	0.84–2.16	0.222
PSA	1.00	1.00–NA	0.710
Presence of a single pelvic node on PSMA	0.38	0.05–1.90	0.283
PSMA positive scan	13.71	4.28–50.40	<0.001

## Data Availability

The datasets used and/or analyzed during the current study are available from the corresponding author on reasonable request.

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
