# Peer review of "PSMA PET-CT Imaging Predicts Treatment Progression in Men with Biochemically Recurrent Prostate Cancer—A Prospective Study of Men with 3 Year Follow Up"

_cancers, 2022, doi:10.3390/cancers14112717_

Round 1

Reviewer 1 Report

The manuscript by Ong et.al. described association between PSMA scan-based treatment decision making and 3 year treatment progression in a cohort of 96 prostate cancer patients with biochemical recurrence. Here are the major and minor concerns.

Major concerns

  1. It is unclear how the results of PSMA scan affected treatment choice. In other words, what were the criteria of choosing options in Table 4 based on PSMA scan findings (Table 2)? The authors should include their treatment algorithms/decision tree which will help external validation of results presented.
  2. What was the follow up protocol used in the study? Which imaging modalities were used? What criteria determined if patients with PSA/imaging progression need treatment progression and which type?
  3. The title of manuscript described PMSA imaging as a “predictor”. There needs to be justification of its predictive role with statistical tests (e.g. logistical regression with PSMA status and occurrence of treatment progression).

Minor concerns

  1. Line 56, PSMA was not targeted by small molecules as Ga or F. It was targeted by its ligand and Ga/F are radio tracers.
  2. Line 160, delete “men” before “3/6”
  3. Table 5, include another row for “progressed to chemotherapy” as mentioned in text.

Author Response

Thank you for your comments. We have detailed our responses to each below.

Comment 1: It is unclear how the results of PSMA scan affected treatment choice. In other words, what were the criteria of choosing options in Table 4 based on PSMA scan findings (Table 2)? The authors should include their treatment algorithms/decision tree which will help external validation of results presented.

Response 1: Thank you for all your comments. We have now included the follow paragraph.

In general, men post radical treatment only were offered salvage prostatectomy or radiotherapy if no targets outside the prostate were seen. Men were offered metastasis directed treatment if feasible or systemic therapy for high volume of disease outside the prostate. We found however, that the majority of men (35 men) opted for surveillance with PSA with a small number of men receiving salvage prostatectomy (5 men) or salvage radiation (11 men). This is likely due to management plans being created through a shared decision-making process based on guidelines, expert opinion, patient preference and availability.

Comment 2: What was the follow up protocol used in the study? Which imaging modalities were used? What criteria determined if patients with PSA/imaging progression need treatment progression and which type?

Response 2: Thank you we have now added the following to explain the follow up protocol used.

“Men received a minimum of 6 monthly PSAs for follow up, with imaging or further presentation to an MDM based on PSA results or clinical concern. The need for treatment progression and type of treatment was discussed in MDM consensus.”

As men underwent a variety of treatment options, the imaging modality and timing of imaging was left to the discretion of their treating clinician and team.

Comment 3: The title of manuscript described PSMA imaging as a “predictor”. There needs to be justification of its predictive role with statistical tests (e.g. logistical regression with PSMA status and occurrence of treatment progression).

Response 3: We have performed an univariate and multivariate analysis of the features that predict a change in management. We found that a positive PSMA scan finding was a significant predictor of the initiation of change in a patients management. These results are presented in the results section and also in a new Table.

Predictors

Odds ratios

Confidence Intervals

P value

Age

1.06

0.97 – 1.16

0.207

Grade group of initial prostate biopsy

1.34

0.84 – 2.16

0.222

PSA

1.00

1.00 – NA

0.710

Presence of a single pelvic node on PSMA

0.38

0.05 – 1.90

0.283

PSMA Positive Scan

13.71

4.28 – 50.40

<0.001

Minor concerns

Comment 4: Line 56, PSMA was not targeted by small molecules as Ga or F. It was targeted by its ligand and Ga/F are radio tracers.

Response 4: Thank you for picking this up. It now reads like this: “When targeted by small ligands with an attached radio-labelled marker such as gallium (68Ga) or Fluorine (18F), PSMA can be combined with PET technology to produce an image that detects the presence of PCa.”

Comment 5: Line 160, delete “men” before “3/6”

Response 5: Thank you this has now been done.

Comment 6: Table 5, include another row for “progressed to chemotherapy” as mentioned in text.

Response 6: Thank you this is now added.

Reviewer 2 Report

This paper reports that for men with BCR, triage with PSMA PET-CT culminates in clinical decisions that have low rates of treatment progression, regardless of pre- or post-BCR treatment. Their results are very interesting because they show that for men with BCR, triage by PSMA PET-CT is associated with a lower rate of treatment progression, regardless of pre- or post-BCR treatment.

However, I believe that further in-depth research is needed before it can be published in Cancers. For a better report, the author should comment on the following points.

1. Why did 44 of the patients with positive PSMA-PET not progress to treatment, especially the 2 patients whose cancer had progressed on imaging and PSA?
The authors claim that overtreatment can be avoided by treatment based on PSMA-PET results, but it seems unnatural that many patients with positive PSMA-PET did not progress in treatment.

2. Is there a difference in recurrence rate between patients with BCR and progression on imaging but negative PSMA-PET and no treatment progression, and patients who received additional treatment at their request?

3. In recent years, with the development of ADT, the number of prostate cancers showing AR-independent growth has been increasing. In other words, AR-independent tumors such as neuroendocrine prostate cancer are known to be PSMA-PET negative, but show very progressive disease despite PSA negativity. What tests (NSE, CT, etc.) should we pay attention to in order not to miss such disease and to make effective use of PSMA-PET?

4 Table 2 and Table 5 should be categorized or otherwise made more readable.

5 If possible, please compare the outcome in the PSMA-PET-positive group with and without additional treatment, and in the PSMA-PET-negative group with and without additional treatment. Also, please perform a multivariate analysis to determine which factors caused the difference.

6 (minor comment)  Within the title, PSMA is misspelled as PMSA. This is a minor but very serious error that needs to be corrected.

Author Response

Thank you for all your comments, below is a detailed reponse to your comments.

Comment 1: Why did 44 of the patients with positive PSMA-PET not progress to treatment, especially the 2 patients whose cancer had progressed on imaging and PSA?
The authors claim that overtreatment can be avoided by treatment based on PSMA-PET results, but it seems unnatural that many patients with positive PSMA-PET did not progress in treatment.

Response 1: Thank you for your comments. In our study we found that 44 men with positive PSMA-PET did not have “treatment progression”. This does NOT mean that they did not get any treatment, but rather that after the initial treatment decision after PSMA PET, they did not have any further additions in treatment over the next 3 years. I hope this makes more sense.

Our statement about overtreatment (and over scanning) comes mainly from our results from men with negative PSMA PET scans who did not have and treatment progression. We state that perhaps for these men, the timing of serial PSMA PET scans and salvage treatment need to be further investigated. A published update of this study with 5-10years of follow up will shed more light on this.

Comment 2. Is there a difference in recurrence rate between patients with BCR and progression on imaging but negative PSMA-PET and no treatment progression, and patients who received additional treatment at their request?

Response 2: This is an interesting concept, unfortunately we are unable to assess an outcome change over time, as these men did not undergo serial PSMA scans so we are unable to assess if these men had further progression after the initiation of treatment. But comparing patients with a Positive PSMA that commenced on another treatment, this was statistically significant (p=0.03) compared to those with a negative PSMA.

Comment 3: In recent years, with the development of ADT, the number of prostate cancers showing AR-independent growth has been increasing. In other words, AR-independent tumors such as neuroendocrine prostate cancer are known to be PSMA-PET negative, but show very progressive disease despite PSA negativity. What tests (NSE, CT, etc.) should we pay attention to in order not to miss such disease and to make effective use of PSMA-PET?

Response 3: Thank you, we have now added the paragraph below in our discussion to touch on this point.

“One thing to consider is the possibility of prostate cancer showing androgen receptor independent growth. With the development of ADT in recent years, the number of these prostate cancers is growing. This has implications for subtypes such as neuroendocrine-differentiated or small cell prostate cancer which are known to be PSMA PET-CT negative but tend to be more lethal.[27] In these cases, clinicians need to be mindful of the negative PSMA PET-CT and consider the utilisation of conventional imaging modalities (CT, WBBS, MRI)[28], novel imaging techniques (Ga-68-DOTATOC PET-CT or DLL3 PET-CT)[29] or measurement of serum neuron specific enolase. [30]”

Comment 4 Table 2 and Table 5 should be categorized or otherwise made more readable.

Response 4:  Thank you. It now looks like below. We hope this is more readable.

Table 2. PSMA PET-CT results.

Positive

63

Negative

33

Prostate lesion only

6

Pelvic lymph nodes

37

Extra-pelvic lymph nodes

14

Bony lesion

12

Distal organ lesion

7

Table 5. Follow up and treatment progression

Follow up

Lost to follow up

4

Median follow up

36 months (IQR 24-40 months)

Treatment progression

No treatment progression

65 patients

Treatment progression

24 patients

Progression on imaging or PSA but no treatment progression

7

Progressed to ADT

15

Progressed to Radiotherapy

6

Progressed to Chemotherapy

10

Comment 5: If possible, please compare the outcome in the PSMA-PET-positive group with and without additional treatment, and in the PSMA-PET-negative group with and without additional treatment. Also, please perform a multivariate analysis to determine which factors caused the difference.

Response 5: We have performed an univariate and multivariate analysis of the features that predict a change in management. We found that a positive PSMA scan finding was a significant predictor of the initiation of change in a patients management. These results are presented in the results section and also in a new Table.

Predictors

Odds ratios

Confidence Intervals

P value

Age

1.06

0.97 – 1.16

0.207

Grade group of initial prostate biopsy

1.34

0.84 – 2.16

0.222

PSA

1.00

1.00 – NA

0.710

Presence of a single pelvic node on PSMA

0.38

0.05 – 1.90

0.283

PSMA Positive Scan

13.71

4.28 – 50.40

<0.001

Comment 6: (minor comment) Within the title, PSMA is misspelled as PMSA. This is a minor but very serious error that needs to be corrected.

Response 6: Thank you for picking up that! A major oversight by us which has now been corrected.

Reviewer 3 Report

  • I suggest providing a more detailed presentation of PSMA , that is expressed in normal, benign and malignant prostatic epithelium and it is not specific to prostatic tissue (doi. 10.23736/S0393-2249.18.03081-3 ; PMID 29664244).In addition , it will be for the benefit of the reader if the author comment on the of the pathophysiology of the PSMA receptor and its role in growth pathways in the cancer cell. In line 65 the authors should add that the positive predictive value of PSMA-PET is likely to be lower in men with solitary lesions, especially in the ribs (doi: 10.2144/fsoa-2021-0035; PMID 34046207).
  • In lines 126-127 what is this standardized protocol? ( i.e. quantity/kg of Gallium injected).
  • The authors should specify that according to the latest Eau guidelines the threshold that best predicts further metastases is a PSA > 0.4 ng/ml and rising.

In addition, in patients with histological findings (after RP), it would be appropriate to report those with poorly differentiated disease, positive surgical margins and lymph node staging.

Author Response

Thank you for your comments, below is a detailed response to your comments.

Comment 1: I suggest providing a more detailed presentation of PSMA , that is expressed in normal, benign and malignant prostatic epithelium and it is not specific to prostatic tissue (doi. 10.23736/S0393-2249.18.03081-3 ; PMID 29664244).In addition , it will be for the benefit of the reader if the author comment on the of the pathophysiology of the PSMA receptor and its role in growth pathways in the cancer cell. In line 65 the authors should add that the positive predictive value of PSMA-PET is likely to be lower in men with solitary lesions, especially in the ribs (doi: 10.2144/fsoa-2021-0035; PMID 34046207).

Response 1: Thank you for your comments and the references. We have now added this paragraph and sentence to include the information you suggested.

PSMA is a non-soluble, type 2 cell surface transmembrane glycoprotein that is expressed on the surface of endothelial cells[1]. Contrary to its name, PSMA is not specific to prostate tissue.[2] In fact, it has been seen in normal, inflammatory, benign and malignant tissues of the body.[2,3] Some studies theorise that PSMA has a pathophysiologic role in carcinogenesis, in particular in regards to folate regulation and tumour specific neovascularisation.[4]  Specific to prostate cancer, PSMA is found in over 90% of PCa cells at expression rates that are 100-1000 times greater than physiological levels especially in higher grade cancers.[5-8] Thus, PSMA has now been targeted for diagnostic imaging and therapeutic advances. Furthermore, the positive predictive value of PSMA PET-CT is likely to be lower in men with solitary lesions, especially in the ribs.[9]

Comment 2: In lines 126-127 what is this standardized protocol? ( i.e. quantity/kg of Gallium injected).

Response 2: We have now added this paragraph:

 “68Gallium-PSMA-11 scanning protocols were similar across sites. PET-CT scanners acquired PET images from thighs to vertex at 60 minutes after administration of 2 MBq/kg body weight ± 5% of 68Gallium-PSMA-11. A low dose non-contrast CT was performed during tidal respiration for attenuation correction and anatomical correlation.”

Comment 3: The authors should specify that according to the latest Eau guidelines the threshold that best predicts further metastases is a PSA > 0.4 ng/ml and rising.

Response 3: Thank you for picking up this update. We have now added this paragraph clarifying the recent changes.

The natural history of men with BCR is highly variable. Currently, post radiation therapy, BCR can be defined as a rise of >2ng/ml above nadir.[10] However, the correct definition of BCR post radical prostatectomy is less clear. The most current EAU-EANM-ESTRO-ESUR-ISUP-SIOG guideline state that after radical prostatectomy, there is no specific PSA threshold defining recurrence. They do however acknowledge that the best threshold for predicting further metastases is a rising PSA of >0.4ng/ml.[11] The AUA-ASTRO-SUO guidelines state that BCR should be defined as 2 consecutive PSA readings >0.2ng/ml that is rising.[12]

Comment 4: In addition, in patients with histological findings (after RP), it would be appropriate to report those with poorly differentiated disease, positive surgical margins and lymph node staging.

Response 4: Thank you we have now included this paragraph in the results.

Of the men who had initial treatment with RP, 30 men had ISUP grade 4 or 5 disease found in their RP specimen (1 patient ISUP score not found). Positive margins and intraductal carcinoma were found in 28 (6 not reported) and 17 (13 not reported) RP specimens respectively. Positive lymph nodes were found in 13 specimens however 23 men did not receive a lymph node dissection. Stage T2, T3a, T3b and T4 were found in 19, 37, 31, 1 men respectively (5 were not reported).

Round 2

Reviewer 2 Report

Thank you for your sincere correction of the points I raised. I consider your paper to be mostly sound and worthy of publication.

I would recommend only one addition from a basic research perspective: on lines 62-64, page 3, you describe the molecular function of PSMA with respect to neovascularisation. There have been several important papers published on this subject in recent years, which should be added to your bibliography (e.g., Watanabe et. al, The Prostate, Dec;81(16):1390-1401, 2021). I believe this addition would further enhance the value of your paper.

Author Response

Thank you for this suggestion and your time for reviewing our manuscript. 

I have now added this publication to our list of references, along with the references below:

Morgantetti G, Ng KL, Samaratunga H, Rhee H, Gobe GC, Wood ST. Prostate specific membrane antigen (PSMA) expression in vena cava tumour thrombi of clear cell renal cell carcinoma suggests a role for PSMA-driven tumour neoangiogenesis. Translational Andrology and Urology. 2019 May;8(Suppl 2):S147.

Gao Y, Zheng H, Li L, Feng M, Chen X, Hao B, Lv Z, Zhou X, Cao Y. Prostate-Specific Membrane Antigen (PSMA) Promotes Angiogenesis of Glioblastoma Through Interacting With ITGB4 and Regulating NF-κB Signaling Pathway. Frontiers in Cell and Developmental Biology. 2021 Mar 4;9:462.

Reviewer 3 Report

The authors answered all comments and suggestions.

Author Response

Thank you for your time given to reviewing our manuscript